# Is a 2D Nanostructured Surface Capable of Changing the Corrosion and Magnetic Properties of an Amorphous Alloy?

**DOI:** 10.3390/ijms241713373

**Published:** 2023-08-29

**Authors:** Irina Kuznetsova, Olga Lebedeva, Dmitry Kultin, Natalia Perova, Konstantin Kalmykov, Petr Chernavskii, Nikolai Perov, Leonid Kustov

**Affiliations:** 1Department of Chemistry, Lomonosov Moscow State University, 119991 Moscow, Russia; kuznetsowair@yandex.ru (I.K.); dkultin@general.chem.msu.ru (D.K.); kbkalmykov@mail.ru (K.K.); chern5@inbox.ru (P.C.); 2Faculty of Physics, Lomonosov Moscow State University, 119991 Moscow, Russia; perova.n@physics.msu.ru (N.P.); perov@magn.ru (N.P.); 3N.D. Zelinsky Institute of Organic Chemistry, Russian Academy of Sciences, Leninsky Prospect 47, 119991 Moscow, Russia; 4Institute of Ecology and Engineering, National Science and Technology University “MISiS”, Leninsky Prospect 4, 119049 Moscow, Russia

**Keywords:** ionic liquid, anodization, nanostructures, magnetic nanoparticles, surface functionalization, magnetic materials, magnetic alloys, corrosion, self-assembly, information storage

## Abstract

In this work, an attempt was made to reveal and explain the influence of the process of formation of 2D nanostructures at the surface of an amorphous alloy (an alloy with the composition Co75Si15Fe5Cr4.5Al0.5 (in at.%) was used for this purpose) on the corrosion and magnetic properties of such an alloy. Two-dimensional nanostructures (nanocells of 100–150 nm in size, which were obtained by anodizing the initial sample in an ionic liquid) are essentially a pattern on the surface of the sample, and they cannot completely cover and block the surface from external effects. It was postulated that the presence of these nanostructures during corrosion and magnetic tests has no significant effect. However, a noticeable inhibition effect was observed during corrosion tests and a less noticeable (but still detectable) effect was observed during magnetic tests. The authors believe that the effect obtained, with a detailed study, can be used to increase the corrosion resistance and to improve the properties of traditional magnetic materials.

## 1. Introduction

Magnetically soft amorphous alloys (AAs) play a key role in the production and conversion of electrical energy [1,2], high-density information storage and spintronic devices [3], biomedicine [4], micro-nanomachine engineering [5], as electrode catalysts [6]. Such alloys have a low coercive force and a relatively high magnetic saturation induction. The low coercivity is due to small magnetocrystalline and magnetoelastic anisotropy values [7]. The addition of chromium [1] and non-metallic components (Si, B) has no negative effect on the magnetic properties and prevents the crystallization of the alloy with time [8,9].

The application of magnetic AAs is accompanied by their corrosion in the atmosphere under the influence of negative environmental factors. These factors accelerate the process of the natural “aging” of alloys and degrade their properties [10]. A lot of attention in the literature is focused on the comparison of the corrosion resistance of crystalline and amorphous alloys [10], as well as the influence of various alloying additives. It has been shown [11] that chromium additives increase the corrosion resistance of AAs and that silicon, on the contrary, reduces the corrosion resistance [12]. The corrosion properties of AAs were mainly examined by using potentiodynamic polarization curves, although the impedance method is more informative in these investigations.

The description and analysis of corrosion processes involves the study of the influence of various factors: the microstructure, surface irregularities, oxide layer composition, and concentration of chloride ions [13]. So the homogeneous structure of AAs contributes to the formation of a uniform protective film, which results in an increase in corrosion resistance [10]. Surface modification can slow down or prevent corrosion processes. Corrosion has a great influence on the magnetic properties of the surface [4]. It has been found [12] that the magnetic stability of cobalt-based AAs decreases after prolonged corrosion in distilled water. The magnetic field can also affect the corrosion properties of the surface. It has been shown that the direction of the applied magnetic field has different effects on the corrosion behavior of a BeCu alloy in a sodium chloride solution [14]. The effect of different magnetic fields on the kinetics of the reduction of magnetite with hydrogen was shown [15].

An optimal combination of corrosion and magnetic properties Is necessary to expand the range of practical applications of cobalt-based AAs. Amorphous alloys exhibit interesting properties due to their domain structure, microstructure contribution, and low anisotropy energy [3]. Excellent soft magnetic properties are demonstrated by Co-based amorphous alloys [16]. Cobalt-based AAs are characterized by a decrease in the maximum values of saturation induction, an increase in the temperature coefficient of magnetic characteristics, high wear resistance, and corrosion resistance. The addition of Fe to cobalt-based amorphous alloys contributes to an increase in saturation magnetization and a decrease in coercive force [17]. To improve the magnetic characteristics and significantly reduce magnetostriction, the partial replacement of iron atoms with aluminum was carried out [18]. Transmission electron microscopy (TEM) indicated that such an improvement in the properties was due to the fine dispersion of (CoFe)SiAl nanoparticles in the amorphous matrix. Nanocrystallization also raised the Curie temperature of the aluminum-containing alloy. Alloys with aluminum have been studied to a low extent.

One method of optimization is to modify the surface with nanostructures. A review [19] showed the role of an ionic liquid (IL) in the formation of nanostructures. Anodizing an alloy in an ionic liquid can lead to the formation of porous oxide, dense oxide film, the appearance of nanotubes, nanocells, nanorolls decorated with nanotubes, and many other forms [19,20,21,22]. The task of using the material determines the type of surface modification. It has been shown that the amorphous alloy Fe70Cr15B15 with a surface modified with hexagonal nanostructures demonstrates a significant anodic shift of the corrosion potential (E_corr_ = + 379 mV) compared to the initial alloy (E_corr_ = −125 mV) [20].

The purpose of this Investigation was to obtain the surface of CoFeCrSiAl AA modified by nanostructures (nanocells) under anodic treatment in BmimNTf_2_ and to compare the corrosion resistance of the modified and original alloys. Comparison of magnetic properties such as domain structure, coercive force, and saturation magnetization for an alloy of this composition has not been carried out before. These properties depend in part on the surface modification. Impedance spectroscopy assists in examining phase interfaces at the electrode surface, which is useful for studying nanoscale oxide layers on the surface of alloys.

## 2. Results and Discussion

### 2.1. Anodic Modification of the Alloy Surface in BmimNTf_2_

Thin oxide films with a developed surface are promising for applications in electrochemical capacitors [23]. The formation of nanocells with four or five walls during the anodizing of silicon in IL with a water content of up to 2% was observed recently [24]. The anodization of aluminum in IL is accompanied by the formation of various nanostructures [25]. These facts made it possible to propose the possibility of producing nanocells on the alloy surface under study. The conditions (current, time) providing the formation of hexagonal nanocells on the AA surface were found (Figure 1). Figure 1 shows SEM images of the surface of the initial Co75Si15Fe5Cr4.5Al0.5 alloy after abrasive treatment and anodizing in IL for 80 s (i = 17.5 mA cm^−2^) before and after corrosion tests in Ringer’s solution. The average cell size was 100–150 nm.

The dynamics of surface changes with the variation of the anodizing time in the IL revealed that the optimal anodizing time for the formation of nanostructures at the same current density (i = 15 mA cm^−2^) ranged from 40 to 300 s (Figure 2). During this time, the cells have time to form completely, but then the process of their “overgrowth” and coverage by oxide occurs (300–1800 s). At longer times, the images become less clear due to the formation of surface oxide, with its film covering the cells.

The elemental composition of the surface after the abrasive treatment (1, “clean” surface), initial (2, natural oxide) and anodizing (3, nanocells on the surface) alloy, and samples after corrosion tests in the Ringer’s solution (see Section 3.4) are shown in Table 1. The data in Table 1 indicate that the surface modification with cells is not accompanied by enrichment with any element. The surface of the initial alloy is coated with natural oxide. Abrasive treatment leads to the removal of natural oxide, but the renewed surface easily interacts with oxygen in air, which dramatically increases the oxygen content compared to the original sample due to the formation of new oxide structures [26].

The role of natural oxide in the formation of nanostructures is discussed in detail earlier [21]. Nanostructures are formed only in the presence of natural oxide of a certain thickness. Anodizing in an ionic liquid can lead to the formation of nanotubes, nanocells, and “nanorolls” depending on the intrinsic properties of the surface oxide. A dense oxide with good adhesion promotes the formation of nanocells. It was found that the natural surface oxide of cobalt has the composition Co_3_O_4_ [27] and does not change the magnetic properties of cobalt itself. Through anodizing in a buffer solution, compact films with good adhesion were obtained on the cobalt surface, the inner layer of which is composed of CoO, and the outer layer consists of Co_3_O_4_ [28].

### 2.2. Electrochemical Corrosion Testing

Electrochemical corrosion of the samples described in Section 3.1 was performed by using potentiodynamic polarization curves in the Ringer’s solution (Figure 3, Table 2) and using the impedance method in the Ringer’s solution and the sodium sulfate solution (Figure 4) [29].

The resulting corrosion potentials and resistances, calculated on the basis of polarization measurements, are shown in Table 2.

Figure 1 and Figure 3, and Table 2 demonstrate that the removal of the natural oxide film via abrasive treatment leads to failure of the alloy’s corrosion resistance. Irregularities appear on the surface of the alloy (Figure 1a,b). After corrosion tests, the AA surface with natural oxide exhibits the least observable changes (Figure 1b,e). The corrosion resistance of the alloy modified with oxide nanostructures is higher than that of the original alloy (Figure 3, Table 2). The anodic modification in IL does not practically change the surface composition. The enrichment of the surface with chromium was not observed (Table 1), allowing us to conclude that the increase in the corrosion resistance is due to the presence of nanostructures. The cobalt content after corrosion decreases for abrasive-treated and modified samples (Table 1). The active dissolution of the alloy in the Ringer’s solution is caused by the formation of a CoCl_4_^2−^ complex. In the presence of a chelating agent, cobalt hydroxide/cobalt oxide can be dissolved, resulting in the occurrence of a transpassive dissolution mechanism [30]. After active dissolution, passivation is observed in both the original and modified samples (Figure 3, curves 2 and 3). The role of chromium as a component that improves the corrosion properties of alloys is not unambiguous. According to [11], chromium significantly improves the corrosion resistance of an iron-based alloy in a hydrochloric acid solution. An investigation of the corrosion product film composition on the surface of a chromium-doped alloy showed that only chromium oxide has protective properties among all the products [31]. For alloys with a higher Co content (68–70 at.%), regardless of the concentration of Cr and Mo, no significant passive behavior was observed [32]. Alloys of cobalt with chromium in the Ringer’s solution showed a tendency toward passivation due to the formation of mixed protective layers of Cr_2_O_3_-CoO with a high stability on their surfaces [33].

Figure 4 shows the impedance spectra of the samples under study in a sodium sulfate solution (Hodograph 2-S and 3-S) and in the Ringer’s solution (Hodograph 1-R and 3-R). For comparison, the impedance spectrum of crystalline cobalt in the Ringer’s solution (Hodograph 4-R) is given.

An equivalent circuit for the process is shown in Figure 4b. The form of the equivalent circuit for all of the studied electrodes and solutions is identical and differs in the resistance value of the solution and the oxide film. The values of the parameters of the equivalent circuit are presented in Table 3.

As the frequency of the AC signal decreases, the impedance spectra (Figure 4) represent a half-circle corresponding to the sum of the Faraday resistance and the volume resistance of the electrolyte. The radius of the “high-frequency arc” of the impedance spectra for the sample with “natural oxide” is comparable to the sample with cells. This section of the diagram is responsible for the outer layers of the sample surface and indicates comparable surface activity of these samples.

The resistance of the Ringer’s solution is higher than that of the sodium sulfate solution as the ion concentration in the sodium sulfate solution is higher. CPE and Rp in the equivalent circuit represent the corrosion layer. The change in CPE and Rp values, as well as their relationship to the value of the corrosion potential, allows the estimation of the protective function of the surface film formed during corrosion. There is a direct correlation between the charge transfer resistance and the protective function of the corrosion layer [34]. The samples show significant resistance to charge transfer (Table 3). For the sample with the nanocell-modified surface, Rp is higher and CPE is approximately the same as for the abrasive-treated sample. Thus, the corrosion layer formed on the nanocell-modified surface is protective [35]. The alloy with the modified surface is more resistant to corrosion than the abrasive-treated one, in agreement with the results of Table 2 showing the shift of the corrosion potential of this sample to the anodic region. The CPE and Rp values for the alloy in the Ringer’s solution are close to those for crystalline cobalt (Figure 4a and Table 3). Particularly high resistance to pitting corrosion induced by chloride ions was found for the cobalt electrode preliminarily passivated at a potential of 0.15 V [36]. The resulting passive film was identified as a thick inner CoO layer and a thin outer Co_3_O_4_ layer. Apparently, the surface modification through anodizing in IL is similar to the effect of pre-passivation.

### 2.3. Magnetic Properties of Studied Samples

The well-defined magnetic properties of AA and their wide application make it useful to investigate the possible correlation of magnetic properties with the corrosion behavior of the alloy, as well as with the presence/absence of nanostructures and oxides on the alloy surface.

The influence of the nanoparticle structure (spherical/hexagonal) on the magnetic properties of cobalt was studied in the literature [37].

The magnetic properties of alloys are sensitive to changes in many factors, including changes in the surface layer structure and composition [38]. Amorphous alloys, due to their structure, have few defects and are capable of magnetization to saturation in magnetic fields of a low strength. Amorphous alloys have a high resistance to corrosion, which can be improved by modifying the surface of the material (Section 2.2).

The coercive force is a structure-sensitive magnetic characteristic of the material, and it can be assumed that modifying the surface with nanostructures changes the value of the coercive force and the domain structure of the surface layer (Table 4).

The coercivity at T = 298 K varies in a small range of values from 0.35 E (Oe) for sample 1 (curve 1_298 K) to 0.45 E (Oe) for sample 3 with nanostructures (curve 3_298 K), which is clearly visible in Figure 5a. After corrosion, the coercivity value increases significantly (Figure 5b), and this can be explained by an increase in the number of defects on the surface [39]. The saturation magnetization is higher for samples 1–3 at 100 K and then decreases at 298 K (Figure 5a).

The coercive force value is lower for samples when magnetized at 100 K, which corresponds to a stronger and lighter magnetization at a lower temperature. Remanence magnetization is stronger for samples after corrosion (Figure 5b).

Figure 6 shows images of the domain structures at room temperature obtained on the samples after and before abrasive treatment (Figure 6a,b) and on the surface modified with nanostructures (Figure 6c). The morphology of the domain structures can be seen to change under the effect of treatment (Figure 6a–c). Various magnetic domain structures can be considered as candidates for use in magnetic storage media [3].

Although nanocells have a noticeable effect on the domain structure at the material surface (Figure 6b,c), the direct effect of modification on the coercive force is worth noting as being very small. The presence of nanostructures on the surface does not significantly change the saturation magnetization and residual magnetization, as illustrated in Figure 7.

The coercive force is sensitive to a deeper surface modification. The thickness of the modified layer is less than 1 µm. After polarization testing in the Ringer’s solution, the surface layer undergoes significant changes with the destruction of the modified layer and the formation of new defects. This is reflected in the magnetic characteristics such as the increase in the value of the coercive force by more than 10 times (Figure 7, Table 4).

The AA has negligible magnetic anisotropy along the surface due to its amorphous nature, which contributes to lower coercive force values for its magnetization. In conclusion, it should be noted that the samples were magnetized in different directions, and the change in the slopes of the hysteresis loops indicates the presence of light and hard axes of magnetization. The shapes and parameters of the hysteresis loops change depending on the orientation of the magnetic field. This indicates that the AA has negligible magnetic anisotropy along the axis, which is directed into the bulk of the sample, where the coercive force values are slightly higher than for measurements along the surface of the samples.

## 3. Materials and Methods

### 3.1. Materials

Ionic liquid 1-butyl-3-methylimidazolium bis(trifluoromethylsulfonyl)imide—BmimNTf_2_ (purity 99.5%) (ABCR GmbH, Germany) was used as an electrolyte to modify the electrode surface. The water content (900 ppm) was determined using Fischer titration using a 899 coulometer (Metrohm, Switzerland). All chemicals in the current study were of high purity and were used without any prior purification. The composition of the alloy under study is Co75Si15Fe5Cr4.5Al0.5 (in at.%). The preparation method of the amorphous alloy is given in Appendix A.

### 3.2. Material Characterization

The amorphous nature of the alloy was confirmed using DSC measurements with a Jupiter STA 449 F1 calorimeter (NETZSCH, Waldkraiburg, Germany). The temperature scanning speed was 20 K min^−1^. X-ray phase analysis was performed using a diffractometer DRON-3 (Cu-Kα). According to DSC (Appendix A) and X-ray diffraction data (Appendix A), the obtained alloy is amorphous. The temperature at the beginning of crystallization was 530 °C. The size of the crystallites was determined (Scherrer equation) from the XRD data and is equal to 1.46 nm.

Surface morphology and composition of the sample were characterized using an EVO-50 Zeiss scanning electron microscope with an EDX (energy dispersive X-ray spectroscopy) analyzer (Zeiss AG, Jena, Germany) and a ZEISS Axio Vert. A1 optical microscope (Zeiss AG, Germany).

### 3.3. Anodizing

All electrochemical experiments were performed at room temperature in air. Anodizing was performed in a three-electrode electrochemical cell with an undivided cathode–anode space under galvanostatic conditions in the current density range 10–18 mA cm^−2^ with a PGSTAT Autolab 302N potentiostat/galvanostat (Metrohm AG, Herisau, Switzerland) in BmimNTf_2_ ionic liquid. The exposure time was 15 to 1800 s. The working and auxiliary electrodes were made of an amorphous magnetically soft cobalt-based alloy (≥70%) with a nominal composition of Co75Si15Fe5Cr4.5Al0.5. The geometric area of the working electrode was 0.4 cm^2^. Silver wire was used as a reference electrode during anodizing.

### 3.4. Corrosion Testing

The corrosion behavior of the initial and modified alloys was studied on the basis of polarization curves and using the impedance method. The standard composition of the Ringer’s solution (1L) is 0.33 g of calcium chloride + 0.3 g of potassium chloride + 8.6 g of sodium chloride, and NaH_2_PO_4_ was used. Potentiodynamic polarization curves in the Ringer’s solution were recorded using a PGSTAT Autolab 302N potentiostat/galvanostat (Metrohm AG, Switzerland) at a sweep rate of 1 mV s^−1^. The reference electrode was a Ag/AgCl one. The auxiliary electrode was a platinum wire. The sample-immersion time corresponded to the measurement time of the polarization curve without preliminary exposure in the medium.

### 3.5. Impedance Response Testing

Impedance spectra were measured in a three-electrode unseparated cell at room temperature in the Ringer’s solution and 1 M Na_2_sO_4_. The reference electrode was an Ag-AgCl reference electrode. The auxiliary electrode was a platinum wire. Measurements were performed by using a P-40X potentiostat with an FRA-24M electrochemical impedance measurement module in the frequency range from 50 kHz to 0.01 Hz with an ac voltage amplitude of 20 mV. The measurements were performed at a potential of −200 mV. The sample-immersion time corresponded to the impedance measurement time without preliminary exposure in the medium.

### 3.6. Magnetic Properties Testing

To register hysteresis loops and observe the magnetic domain structure using the MOKE method, the magneto-optical Kerr microscope Evico magnetics GmbH, Germany, based on a Carl Zeiss polarization microscope, was employed. All magneto-optical measurements were performed at room temperature and at a high resolution with standard magnetizing coils without cores. A ZEISS EC Epiplan-NEOFLUAR 20x/0.50 Pol focusing lens was used.

Measurements of volumetric magnetic properties were carried out with a LakeShore magnetometer VSM (vibrating sample magnetometer), model 7407 (USA), at temperatures from 100 to 450 K in fields up to 16 kE. The mass of the samples was determined using analytical balances Rawdag (Germany) with a resolution of 0.01 mg. The magnetometer was calibrated using a nickel standard with a magnetic moment of 6.92 emu in the field of 5 kE.

## 4. Conclusions

Functional modification of the Co75Si15Fe5Cr4.5Al0.5 AA surface was carried out via nanostructures. Nanostructures—hexagonal cells in the cross-section of 100–150 nm—were obtained by anodizing them in ionic liquid BmimNTf_2_.

Corrosion resistance of the initial alloy, after abrasive treatment, and anodizing AAs were investigated using the polarization curve method and impedance spectroscopy. Both methods showed convergent results. The corrosion resistance of the alloy modified with oxide nanostructures is higher than that of the original alloy. For the sample with the nanocell-modified surface, Rp is higher and CPE is approximately the same as for the abrasive-treated sample. Thus, the corrosion layer formed on the nanocell-modified surface is protective. An equivalent scheme describing the process is proposed. The view of the equivalent scheme for all investigated solutions and samples is identical and differs in the value of the resistance of the solution and the oxide film.

The magnetic properties and corrosion resistance of the modified and initial alloys were compared. A comparison of such magnetic properties as the domain structure, coercive force, and saturation magnetization for an alloy of this composition has not been carried out before. It has been shown that these properties are affected by the surface modification. Although nanocells have a noticeable effect on the domain structure at the material surface, the direct effect of modification on the coercive force is relatively insignificant. The presence of nanostructures on the surface does not significantly change the saturation magnetization and residual magnetization.

The results allow the prediction of a possible field of application of a nanocell-modified magnetic alloy in the field of information storage on magnetic materials. Two-dimensional nanostructures (nanocells) should allow one to easily record information (domain carriers) and at the same time to keep it without losses for a longer time compared with usual (traditional) carriers.

## Figures and Tables

**Figure 1 ijms-24-13373-f001:**
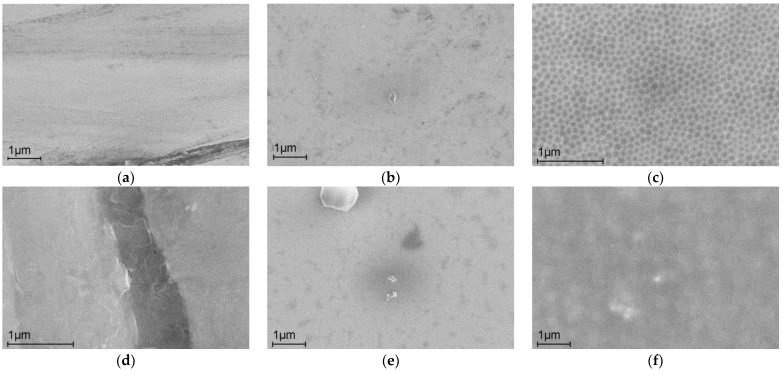
SEM images of the surface: (**a**) after abrasive treatment, (**b**) the initial alloy, (**c**) anodizing in IL for 80 s (i = 17.5 mA cm^−2^), and (**d**–**f**) after corrosion tests.

**Figure 2 ijms-24-13373-f002:**
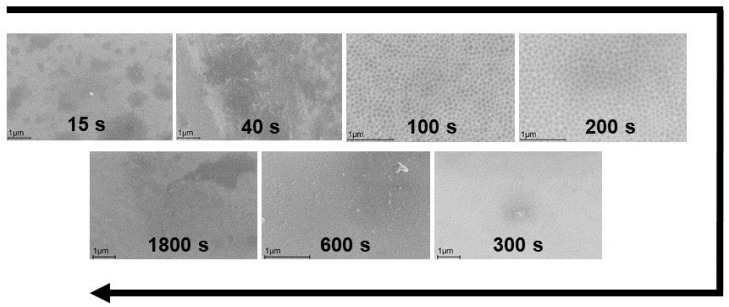
Dynamics of the morphology of the alloy surface during anodizing in IL (i = 15 mA cm^−2^).

**Figure 3 ijms-24-13373-f003:**
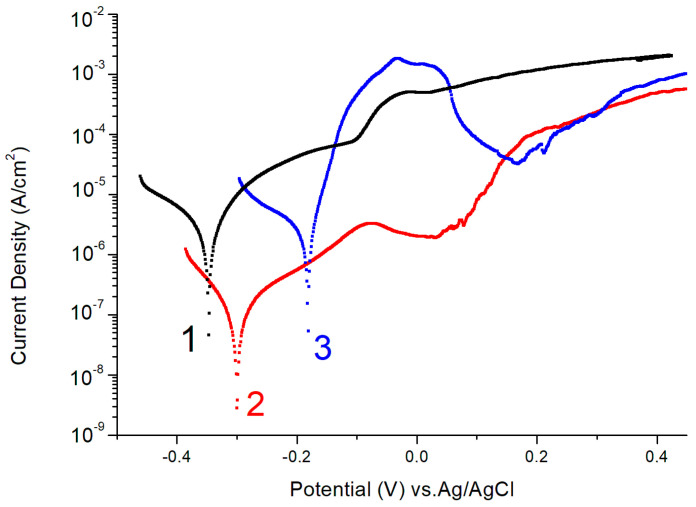
Linear polarization curves (scanning rate 1 mV·s⁻¹ in Riger’s solution) for samples 1–3 of amorphous Co75Si15Fe5Cr4.5Al0.5 alloy. All measurements were carried out using both anodic and cathodic regions. A more detailed description of the samples is given in the text and Appendix B.

**Figure 4 ijms-24-13373-f004:**
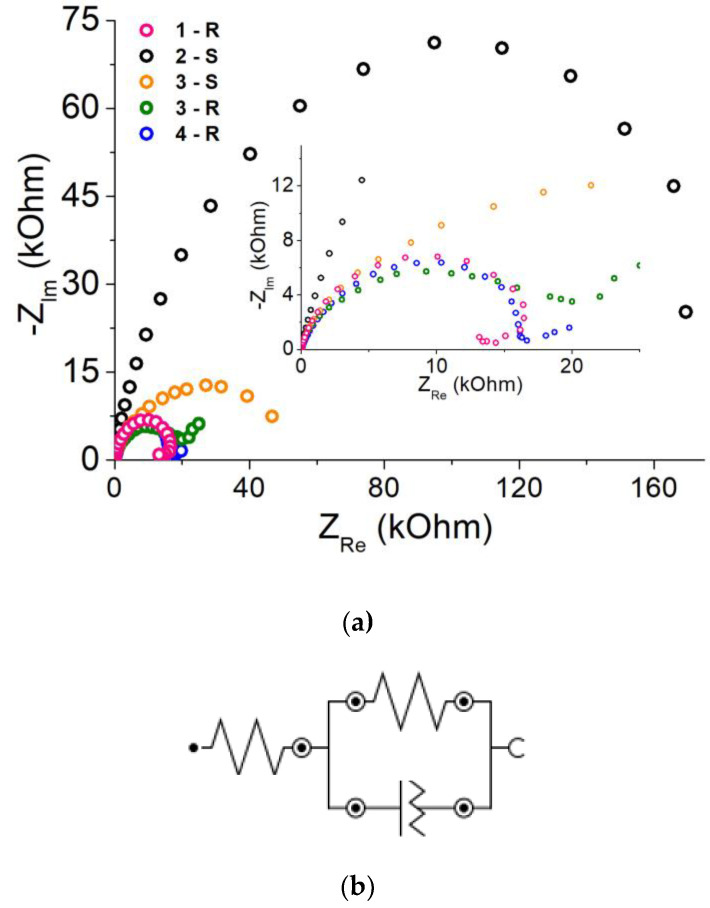
Electrochemical impedance response for electrodes: (1) after abrasive treatment, (2) initial, (3) modified, (4) crystal Co electrode (indexes denote: S–solution of Na_2_SO_4_, R–Ringer’s solution); insert: fragments of hodographs in the ZRe area 0–25 kOhm cm^2^; (**b**) the equivalent scheme. A more detailed description of the samples is given in the text and Appendix B.

**Figure 5 ijms-24-13373-f005:**
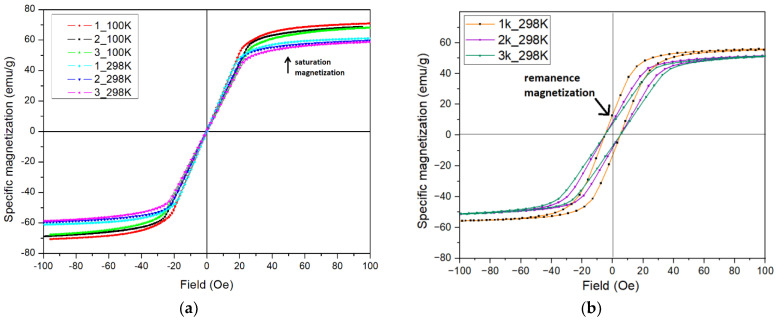
Magnetization curves of AA samples (**a**) before corrosion tests at T = 298 K and T = 100 K and (**b**) after corrosion testing at T = 298 K. A more detailed description of the samples is given in the text and Appendix B.

**Figure 6 ijms-24-13373-f006:**
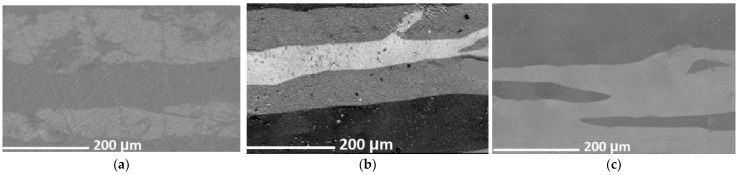
Domain structures of the surface of the samples (**a**)—(1); (**b**)—(2); (**c**)—(3) during magnetization along one axis obtained at T = 298 K.

**Figure 7 ijms-24-13373-f007:**
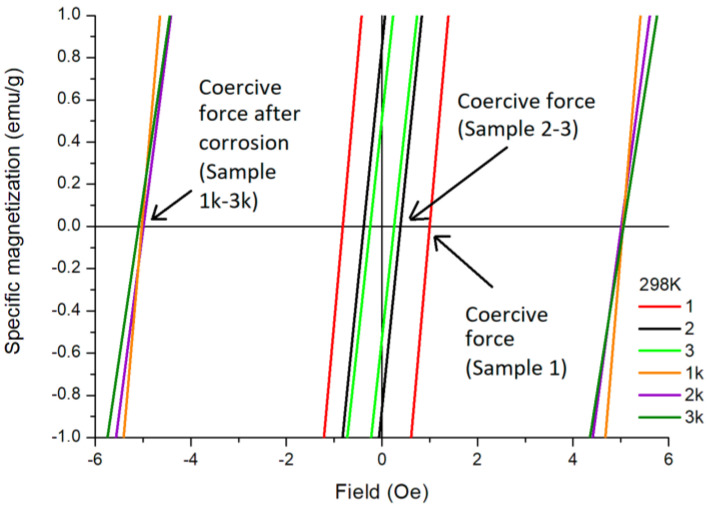
The central parts of the hysteresis loops of samples at T = 298 K. A more detailed description of the samples is given in the text and Appendix B.

**Table 1 ijms-24-13373-t001:** The elemental composition of the alloy before and after corrosion tests.

Sample ^1^	The Content of Elements before Corrosion Tests (at. %)					
	Co	Si	Fe	Cr	Al	O
(1)	72.55 ± 2.40	14.06 ± 0.90	4.73 ± 0.37	3.94 ± 0.17	0.30 ± 0.22	4.42 ± 3.44
(2)	75.10 ± 0.49	14.27 ± 0.39	4.63 ± 0.37	4.19 ± 0.17	0.17 ± 0.16	1.65 ± 0.70
(3)	75.18 ± 0.49	14.54 ± 0.40	4.98 ± 0.25	4.21 ± 0.29	0.29 ± 0.15	0.81 ± 0.12
	**The Content of Elements after Corrosion Tests (at. %)**					
(1k)	70.65 ± 0.91	14.64 ± 0.61	4.76 ± 0.34	3.78 ± 0.24	0.34 ± 0.30	5.82 ± 1.07
(2k)	75.30 ± 1.12	14.37 ± 0.57	4.75 ± 0.24	4.16 ± 0.25	0.30 ± 0.27	1.13 ± 0.73
(3k)	71.39 ± 1.54	14.12 ± 0.28	4.85 ± 0.45	4.03 ± 0.26	0.23 ± 0.18	5.37 ± 1.47

^1^ The K index in the sample name means that the sample was studied after corrosion tests. A more detailed description of the samples is given in the text and Appendix B.

**Table 2 ijms-24-13373-t002:** Experimental and calculated corrosion potentials and polarization resistance in the Ringer’s solution: (1) alloy after abrasive treatment, (2) initial alloy, and (3) modified alloy.

Sample	E_corr exp_ (mV)	E_corr calc_ (mV)	PR × 10^5^ (Ohm)
(1)	−348	−372	0.06
(2)	−319	−322	1.76
(3)	−189	−188	0.04

**Table 3 ijms-24-13373-t003:** The values of the parameters of the equivalent circuit of the studied samples: Rs is the electrolyte resistance; Rp, CPE are the metal charge transfer resistance and double layer capacitance; N is the degree for calculations of CPE.

Sample ^1^	Environment	Rs/Ohm cm^2^	Rp/Ohm cm^2^	CPE/Ohm^−1^ cm^−2^ c^N^	N
(1)	Ringer’s solution	20.4 ± 2%	15455 ± 2%	1.88 × 10–5 ± 3.4%	0.88 ± 0.7%
(2)	1 M Na_2_SO_4_	8.9 ± 3%	173120 ± 3%	6.34 × 10–6 ± 2.1%	0.87 ± 0.4%
(3)	1 M Na_2_SO_4_	10.3 ± 7%	33546 ± 10%	5.09 × 10–5 ± 5.2%	0.84 ± 1.5%
(3)	Ringer’s solution	26.3 ± 4%	19669 ± 5%	3.34 × 10–5 ± 5.8%	0.83 ± 1.3%
(4)	Ringer’s solution	34.0 ± 2%	17117 ± 2%	1.01 × 10–5 ± 2.5%	0.79 ± 0.4%

^1^ A more detailed description of the samples is given in the text and Appendix B.

**Table 4 ijms-24-13373-t004:** The values of the coercive force, saturation, and remanence magnetization obtained for the initial samples and after corrosion testing (Section 2.2) at room temperature (298 K) and T = 100 K.

Conditions	Sample ^1^	Coercivity at 298 K (Oe)	Coercivity at 100 K (Oe)	Saturation Magnetization (Ms) at 298 K (emu/g)	Remanence Magnetization (Mr) at 298 K (emu/g)
Before corrosion testing	(1)	0.35	0.16	61	0.9
	(2)	0.38	0.08	60	0.4
	(3)	0.45	0.01	59	1.3
After corrosion testing	(1k)	5.00	4.70	51	11.5
	(2k)	4.95	4.50	48	9.0
	(3k)	5.10	4.80	47	8.0

^1^ The K index in the sample name means that the sample was studied after corrosion tests. A more detailed description of the samples is given in the text and Appendix B.

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
