# Peer review of "Is a 2D Nanostructured Surface Capable of Changing the Corrosion and Magnetic Properties of an Amorphous Alloy?"

_ijms, 2023, doi:10.3390/ijms241713373_

Round 1

Reviewer 1 Report

The authors of this paper attempted to coat a 2D nanostructured film on a Co75Si15Fe5Cr4.5Al0.5 alloy material and tested the effect of this coating on the corrosion and magnetic properties of the material using a number of tools including electrochemistry, SEM, and VSM. The current status of the paper is not yet sufficient for publication in the International Journal of molecular Science.  There are some commnets:

1. The authors' characterization of the curves in Fig. 3 is incomplete; the polarization curves involve corrosion currents in addition to corrosion voltages, with the values of the voltages reflecting more of a thermodynamic or corrosion tendency and the currents reflecting more of a kinetic character. From the results of Fig. 3, although the potential of curve 2 is more negative than 3, its corrosion current or corrosion rate is obviously smaller than that of curve 3, which indicates that the original state of the sample corrodes at a slower rate, what do the authors think about this?

2. The author uses "crystalline cobalt" as a comparison in the description on page 6, line 221, but why not use the crystalline state of AA material for comparison? This would at least maintain consistency in composition, so please explain.

3. The authors should explain more about the object of study, for example, as an amorphous material, its critical crystallization size is very important, which is related to the test used by the authors, the test area used by EIS is 0.4cm2, in this range of the samples exposed to the amorphous phase are all amorphous? Why?

4. As can be seen from Table 1, the authors used different sample compositions for the polarization curve measurements, can this difference explain the differences presented in the polarization curve results?

5. what are the corrosion products after the corrosion experiments? Why these substances?

6. The authors should describe the experimental procedure in more detail to clearly correspond to the state of the samples, e.g., sample immersion time, the numbering of the samples for the EIS test is very confusing and finds no correspondence with the polarization curves, please ask the authors to renumber the samples.

7. The necessity of some references is doubtful, for example, in line 90 the author mentions literature 17, but literature 17 does not appear at all Co75Si15Fe5Cr4.5Al0.5 material, but only also mentions X-ray and other testing methods.

8. The authors need a more in-depth explanation of the differences in material properties, especially with regard to the magnetic properties. The current explanation does not effectively explain the reasons for the differences in properties; are the differences related only to the nanomembranes? Or is it also related to the structure of the membrane? Some key parameters, such as the film thickness, the state of the film on the surface, and the presence of holes, are not discussed in detail.

The English writting is clear and plainly.

Author Response

The authors of this paper attempted to coat a 2D nanostructured film on a Co75Si15Fe5Cr4.5Al0.5 alloy material and tested the effect of this coating on the corrosion and magnetic properties of the material using a number of tools including electrochemistry, SEM, and VSM. The current status of the paper is not yet sufficient for publication in the International Journal of Molecular Science.  There are some comments:

We thank the Reviewer for in-depth study of our manuscript and valuable comments. We hope that we have answered all the comments exhaustively.

  1. The authors' characterization of the curves in Fig. 3 is incomplete; the polarization curves involve corrosion currents in addition to corrosion voltages, with the values of the voltages reflecting more of a thermodynamic or corrosion tendency and the currents reflecting more of a kinetic character. From the results of Fig. 3, although the potential of curve 2 is more negative than 3, its corrosion current or corrosion rate is obviously smaller than that of curve 3, which indicates that the original state of the sample corrodes at a slower rate, what do the authors think about this?

Response: The graph in Figure 3 is given for the visible surface of the electrodes under study, which was everywhere the same with the surface value of 0.4 cm2. However, the value of the electrochemically active area (ECSA) of the surface should be taken into account. We evaluated ECSA based on the capacity of the double layer by cyclic voltammetry (CV) at scanning rates of 20, 40, 60, 80, 100 mV/s in a sodium sulfate solution and found the value of the capacity of the double electric layer according to the standard technique [Connor, P.; Schuch, J.; Kaiser, B.; Jaegermann, W. The Determination of Electrochemical Active Surface Area and Specific Capacity Revisited for the System MnO x as an Oxygen Evolution Catalyst. Z. Phys. Chem. 2020, 234, 979–994, doi:10.1515/zpch-2019-1514. or Łukaszewski, M.; Soszko, M.; Czerwiński, A. Electrochemical Methods of Real Surface Area Determination of Noble Metal Electrodes – an Overview. International Journal of Electrochemical Science 2016, 11, 4442–4469, doi:10.20964/2016.06.71.]. The lowest value of the capacitance (surface) is characterized by the sample 2 with natural oxide (10.5 µF). The polished sample 1 has a capacity of 73.1 µF, and the sample with nanocells 3 has an intermediate capacity value of 25.6 µF. If the electrochemically active area is taken into account, then the value of the corrosion current density for the sample with nanocells is comparable to the current density of the sample with natural oxide. For this reason, the corrosion potential was chosen to characterize the corrosion resistance, which more clearly demonstrates the higher thermodynamic stability of the sample with nanocells. We did not include these results in the publication, since further experimental research and confirmation by other methods, for example, CO adsorption, are required, and in order not to overload the article. The dependence of the double layer capacity value is presented below:

  1. The author uses "crystalline cobalt" as a comparison in the description on page 6, line 221, but why not use the crystalline state of AA material for comparison? This would at least maintain consistency in composition, so please explain.

Response: Thank you for the excellent question. We also initially planned to prepare a crystalline sample corresponding to AA. Crystallisation of an amorphous alloy at a high temperature is accompanied either by the formation of nanocrystals or by the formation of a multiphase structure in the alloy. In fact, it turned out that it was not possible to prepare such a sample, due to the presence of Si in the composition. When annealed (700°C), the sample gives two phases.

This follows from the phase diagrams and is also clearly visible from the data of DSC and XRD (Supplementary Materials). We consulted with our colleagues in materials science and the best choice was crystalline Co.

Comparison of the properties of an amorphous alloy and pure cobalt is justified due to the isostructure of a cobalt-based solid solution in an alloy of this composition and pure cobalt at temperatures close to room temperature. In both cases, cobalt crystallizes in a hexagonal modification (Mg structural type). It is incorrect to compare the properties of an amorphous alloy and an equilibrium crystalline alloy of the same composition due to the presence of the second phase – Co2Si intermetallic compound. Comparison with a solid silicon solution in the Co(rt) phase is also not possible due to the narrow homogeneity region, which passes into the two-phase region when the alloy is cooled. Probably, additional alloying components shift the boundaries of the silicon solution in cobalt in the way shown by the dotted line. Then the peak temperatures on the DSC curves will be shifted, as seen on the DSC curve.

We are pleased that the Reviewer has the same thought as the authors, thank you for a deep reading of our manuscript and hope that we have answered the question satisfactorily.

  1. The authors should explain more about the object of study, for example, as an amorphous material, its critical crystallization size is very important, which is related to the test used by the authors, the test area used by EIS is 0.4cm2, in this range of the samples exposed to the amorphous phase are all amorphous? Why?

Response: We have added the missing data to Section 3, Materials and Methods, and to Supplementary Materials. DSC and XRD (Supplementary Materials) data were added. From the point of view of the amorphous nature, essentially the same sample was used everywhere, including the composition and method of preparation (added in Supplementary Materials). The difference was in its surface area (natural oxide, bare surface or nanostructures). The value of 0.4 cm2 is the geometric (visible) surface area (this clarification has been added). The size of the crystallites were determined (Scherrer equation) from the XRD data and is equal to 1.46 nm.

  1. As can be seen from Table 1, the authors used different sample compositions for the polarization curve measurements, can this difference explain the differences presented in the polarization curve results?

Response: As it was explained in the answer to the previous question, the sample (if it is considered as a substrate) was the same, the difference appeared only on the pretreated surface (grinding or anodizing) and only because of the presence of surface 2D-nanostructures there is a difference in polarization curves.

  1. What are the corrosion products after the corrosion experiments? Why these substances?

Response: Cobalt oxides are the main surface corrosion products, which follows from the data presented in Table 1. At the same time: (a) no other elements (for example, Cl) were detected; (b) the Co content in clean (1) and anodized (3) samples decreases.

  1. The authors should describe the experimental procedure in more detail to clearly correspond to the state of the samples, e.g., sample immersion time, the numbering of the samples for the EIS test is very confusing and finds no correspondence with the polarization curves, please ask the authors to renumber the samples.

Response: Thank you, this was corrected. The parts 3.4 and 3.5 have been expanded. The numbering of the samples has been revised for the greater convenience of readers. A new section has been added to explain the numbering of all samples: Appendix A.

  1. The necessity of some references is doubtful, for example, in line 90 the author mentions literature 17, but literature 17 does not appear at all Co75Si15Fe5Cr4.5Al0.5 material, but only also mentions X-ray and other testing methods.

Response: This was corrected. Here the reference and phrase have been removed.

  1. The authors need a more in-depth explanation of the differences in material properties, especially with regard to the magnetic properties. The current explanation does not effectively explain the reasons for the differences in properties; are the differences related only to the nanomembranes? Or is it also related to the structure of the membrane? Some key parameters, such as the film thickness, the state of the film on the surface, and the presence of holes, are not discussed in detail.

Response: Thank you for this important comment. Actually, we have just started a systematic study of the effects of the nanocells formation on the magnetic properties of different alloys and pure metals. We found that the size of the cells ranges from 10 to 100 nm. The depth of nanocells is also variable. It looks like the formation of nanocells affects the size of domains. So we need to accumulate more data to give a satisfactory explanation of the effect. However, this effect is reproducible.

Reviewer 2 Report

Thank you for the opportunity to review this interesting paper.

Some of the figure pictures are rather unclear (Figures 1, 2).

Is “Ringer’s solution” the most appropriate term for the solution used in this work?

Please use either full-stops (periods) or Commas – not a mixture of both – to represent decimal points.

According to the Instructions for Authors, “SI Units (International System of Units) should be used. Imperial, US customary and other units should be converted to SI units whenever possible”.

The paper gives the impression of trying to cover quite a lot of ground and, as a result, is not a "standalone” piece of work. For example, several Methodology aspects are referenced as described in other published work. There are many references to other publications given in the Results and Discussion section – to confirm the findings? The explanation of the results and conclusions could be clearer.

Figure 4a: could use kOhms units and use slightly less space on zeros?

Figure 4b: could explain the equivalent circuit better using a little more detail.

Please double-check the authors’ email addresses for errors – just in case.

Please format the manuscript exactly according to the journal guidelines.

Generally, the standard of the written English is very good.

Just a couple of minor spelling/typos were noticed: lines 19, 187.

Author Response

Thank you for the opportunity to review this interesting paper.

We thank the Reviewer for the great work with our manuscript, especially for the careful reading. Many shortcomings were previously hidden from us, but now we hope that everything is corrected.

Some of the figure pictures are rather unclear (Figures 1, 2).

Response: Thank you, this was corrected. In Figure 1, the images were replaced with those of the maximum resolution. In Figure 2, in addition, an error with the image of (300 s) was corrected.

Is “Ringer’s solution” the most appropriate term for the solution used in this work?

Response: Corrosion is usually investigated in solutions of sulfuric acid, sodium sulfate or chloride [31], distilled water [12], the Ringer's solution [32]. According to [13], description and analysis of corrosion processes involves the study of the influence of various factors: microstructure, surface irregularities, oxide layer composition, and concentration of chloride ions. It was the reason for choosing the Ringer's solution. The standard composition of the Ringer's solution (1 L) is: 0.33 g of calcium chloride + 0.3 g of potassium chloride + 8.6 g of sodium chloride and NaH2PO4. The composition of the solution used has now been added to Part 3.4 “Corrosion testing”.

Please use either full-stops (periods) or Commas – not a mixture of both – to represent decimal points.

Response: Yes, we found and corrected such an error in Table 2.

According to the Instructions for Authors, “SI Units (International System of Units) should be used. Imperial, US customary and other units should be converted to SI units whenever possible”.

Response: Thank you, we double-checked the use of SI units and don't have any contradictions right now.

The paper gives the impression of trying to cover quite a lot of ground and, as a result, is not a "standalone” piece of work. For example, several Methodology aspects are referenced as described in other published work. There are many references to other publications given in the Results and Discussion section – to confirm the findings? The explanation of the results and conclusions could be clearer.

Response: Thanks for the comment. The paper attempts to consider the effect of pretreatment on two independent functions: corrosion and magnetic properties. This leads to quoting a large number of sources. Much attention has been paid to the corrosion resistance of cobalt-based amorphous alloys in the literature, but there is no consensus on the effect of alloying additives (Cr), which we tried to demonstrate [11, 31-33]. The alloy under study showed an increase in the corrosion resistance due to the formation of nanostructures, and not the enrichment of the surface with chromium, as is often noted.

Figure 4a: could use kOhms units and use slightly less space on zeros?

Response: OK, this was corrected.

Figure 4b: could explain the equivalent circuit better using a little more detail.

Response: OK, it was done. We made a minor addition right after Figure 4.

Please double-check the authors’ email addresses for errors – just in case.

Response: Thank you for your reminder. We double-checked and put everything in order.

Please format the manuscript exactly according to the journal guidelines.

Response: OK, it was done. We hope that now the manuscript is in exact accordance with the template and Instructions for Authors.

Comments on the Quality of English Language. Generally, the standard of the written English is very good. Just a couple of minor spelling/typos were noticed: lines 19, 187.

Response: Thank you. We check the use of English and corrected typos.

Reviewer 3 Report

In this paper, the authors investigated the magnetic and corrosion properties of the amorphous Co75Si15Fe5Cr4.5Al0.5 alloy with the surface modification. This article is sufficiently novel and interesting to warrant publication. However, I would like to make a few comments/suggestions:

-Please give more information about the investigated CoFeCrSiAl alloy in the introduction section. Why was this alloy composition chosen for the study? What is the point(s) in progress rather than that in the published report?

-I advise the authors to enrich the Introduction section by adding references on magnetic amorphous alloys.

-In abstract and experimental section, the composition of the alloy should be specified as at.% or wt.%.

- In Materials and Methods: What is the production process of the studied sample? As ref.17 provides information for Fe70Cr15B15 alloy, it would be useful to provide more information on the production process of CoFeCrSiAl amorphous alloy.

-The amorphous structure of the produced CoFeCrSiAl alloy can be shown by adding XRD and DSC graphs to the manuscript.

-Line 155: please check the written values which are different than figure 2. “…ranged from 40 to 300 s (Figure 2).”

-In Table 3: please give information for N parameter.

-In Table 4: please add the values of saturation magnetization (Ms) and remanence magnetization (Mr). Also, the obtained magnetization values could be compared with similar alloy values in the literature.

-The Conclusions section could be revised, and obtained results could be given rather than common sentences.

Author Response

In this paper, the authors investigated the magnetic and corrosion properties of the amorphous Co75Si15Fe5Cr4.5Al0.5 alloy with the surface modification. This article is sufficiently novel and interesting to warrant publication. However, I would like to make a few comments/suggestions:

We thank the Reviewer for the high assessment of our work and valuable comments. We hope that the changes in the manuscript and the answers provided below are satisfactory.

-Please give more information about the investigated CoFeCrSiAl alloy in the introduction section. Why was this alloy composition chosen for the study? What is the point(s) in progress rather than that in the published report?

Response: The general criteria for choosing this alloy are based both on the general advantages of soft magnetic amorphous alloys (reduction of the maximum values of saturation induction, increase in the temperature coefficient of magnetic characteristics, high wear resistance and corrosion resistance) and particular advantages. It is believed that the addition of Fe to cobalt-based amorphous alloys contributes to an increase in saturation magnetization and a decrease in the coercive force [17]. The addition of chromium leads to an increase in the corrosion resistance [31]. Silicon contributes to the stabilization of the amorphous state [12]. To improve the magnetic characteristics and significantly reduce magnetostriction, partial replacement of iron atoms with aluminum was carried out [18]. In addition, in such alloys, due to the distribution of nanoscale crystalline granules in an amorphous matrix, very high values of magnetic permeability are achieved, a small value of the coercive force, and the saturation induction (Bs) is 0.8...1.2 T. Alloys with aluminum have been studied to a low extent. The Introduction section has been supplemented with a text that also serves as an answer to the following question.

-I advise the authors to enrich the Introduction section by adding references on magnetic amorphous alloys.

Response: ОК, it was added in Introduction section.

-In abstract and experimental section, the composition of the alloy should be specified as at.% or wt.%.

Response: ОК, it was added in relevant sections.

- In Materials and Methods: What is the production process of the studied sample? As ref.17 provides information for Fe70Cr15B15 alloy, it would be useful to provide more information on the production process of CoFeCrSiAl amorphous alloy.

Response: We agree that the reference was not given according to a completely correct methodology. This has now been corrected. Now, the method of preparation of the alloy is given in Supplementary Materials.

-The amorphous structure of the produced CoFeCrSiAl alloy can be shown by adding XRD and DSC graphs to the manuscript.

Response: These data were added in Supplementary Materials (Figures S1,S2,S3)

-Line 155: please check the written values which are different than figure 2. “…ranged from 40 to 300 s (Figure 2).”

Response: It was corrected in Section 3.3. Anodizing: the range was corrected from 15 to 1800 s (as in Figure 2). Here we are talking about the interval of formation of nanostructures, which was 40-300 s (as can be seen for 100, 200 and 300 s (300 s: already nanocells begin to close with oxide). Explanations have been added to the text.

-In Table 3: please give information for N parameter.

Response: N is the degree for calculations of CPE. This value is obtained from the data on the phase angle of inclination N = arcsin (ZIm/ZRe). We can, of course, indicate that this is a dimensionless quantity, but in the dimension of [Ohms/Ohms], but we believe that it is better to leave it as it is, while the explanation in a caption to the table was added.

-In Table 4: please add the values of saturation magnetization (Ms) and remanence magnetization (Mr). Also, the obtained magnetization values could be compared with similar alloy values in the literature.

Response: The required values have been added to Table 4. The magnetic characteristics can vary significantly depending on the temperature and the degree of crystallization [16]. For this reason, we do not give a comparison in the article. We compare our results with Co72-xFexB28-y (where B includes non-magnetic elements such as boron, silicon, etc. x varies from 4 to 5%, and y varies from 0 to 2%). The coercive force (298 K) is 0.35-0.45 Oe (Table 4), which is lower than the reported value [16].

-The Conclusions section could be revised, and obtained results could be given rather than common sentences.

Response: It was corrected. The part Conclusions has been expanded.

Round 2

Reviewer 1 Report

The authors' responses regarding COMMENTS are in line with expectations, especially in the appendix where more results are given, which better supports the arguments in the manuscript and improves the quality of the paper. Therefore, the paper is publishable now.

The English writing of the manuscript is easy to read and understand.

Reviewer 3 Report

I have gone through the manuscript and am satisfied with the revision. I am pleased to recommend the paper for publication in the journal, congratulations.